# Preparation of Highly Porous PAN-LATP Membranes as Separators for Lithium Ion Batteries

**DOI:** 10.3390/nano9111581

**Published:** 2019-11-07

**Authors:** Jagdeep Mohanta, O Hyeon Kwon, Jong Hyeok Choi, Yeo-Myeong Yun, Jae-Kwang Kim, Sang Mun Jeong

**Affiliations:** 1Department of Chemical Engineering, Chungbuk National University,1 Chungdae-ro, Seowon-gu, Cheongju, Chungbuk 28644, Korea; jagdeepmohanta@gmail.com (J.M.); qweaq1@naver.com (J.H.C.); 2Department of Solar & Energy Engineering, Cheongju University, Cheongju, Chungbuk 28503, Korea; ohhyeon2940@naver.com; 3Department of Environmental Engineering, Chungbuk National University, 1 Chungdae-ro, Seowon-gu, Cheongju, Chungbuk 28644, Korea; ymyun@cbnu.ac.kr

**Keywords:** PAN, LATP, electrospinning, separator, lithium ion battery, electrochemical characteristics

## Abstract

Separators are a vital component to ensure the safety of lithium-ion batteries. However, the commercial separators employed in lithium ion batteries are inefficient due to their low porosity. In the present study, a simple electrospinning technique is adopted to prepare highly porous polyacrylonitrile (PAN)-based membranes with a higher concentration of lithium aluminum titanium phosphate (LATP) ceramic particles, as a viable alternative to the commercialized separators used in lithium ion batteries. The effect of the LATP particles on the morphology of the porous membranes is demonstrated through Field emission scattering electron microscopy. X-ray diffraction and Fourier transform infrared spectra studies suitably demonstrate the mixing of PAN and LATP particles in the polymer matrix. PAN with 30 wt% LATP (P-L30) exhibits an enhanced porosity of 90% and is more thermally stable, with the highest electrolyte uptake among all the prepared membranes. Due to better electrolyte uptake, the P-L30 membrane demonstrates an improved ionic conductivity of 1.7 mS/cm. A coin cell prepared with a P-L30 membrane and a LiFePO4 cathode demonstrates the highest discharge capacity of 158 mAh/g at 0.5C rate. The coin cell with the P-L30 membrane also displays good cycling stability by retaining 87% of the initial discharge capacity after 200 cycles of charging and discharging at 0.5C rate.

## 1. Introduction

A negligible memory effect, superior energy density, long cycle life, and environmental friendliness are some of the important characteristics that have allowed lithium ion batteries to dominate the commercial secondary battery market over the last three decades. Owing to their incredible advantages, lithium ion batteries are employed in various fields from smart phones to large electric vehicles [1,2,3,4,5,6,7,8,9]. In general, electrodes (i.e., anode and cathode), electrolytes, and separators constitute a lithium ion battery. Among these, the separator is an intrinsic component, as the safety of the lithium ion battery relies on its properties. The separator is the component that keeps the two electrodes apart, as their contact may lead to the battery short-circuiting. Apart from this, the separator is very valuable in terms of assisting ion transportation during both charging and discharging of the lithium ion battery [10,11,12,13,14,15,16]. Enhanced porosity, good electrolyte uptake, and better thermal stability are some of the essential features that help polymer membranes qualify as a competent separator for lithium ion batteries [17]. There are many methods for fabricating separators for lithium ion batteries, such as solvent casting, non-solvent-induced phase separation, electrospinning, the dry and wet method, and thermally induced phase separation. Of these, electrospinning is the most favorable technique for preparing separators due to its simple process and its ability to generate highly porous polymer membranes [18].

Until now, the commercial separator primarily employed in large-scale battery production has consisted of polyethylene and/or polypropylene because of the benefits these materials have when used as a separator, such as their economical cost, high electrochemical sturdiness, superior mechanical strength, and inbuilt shutdown feature, which prevent small functions in the case of increased temperatures. These features make polyolefin-based separators the primary choice for battery manufacturers. Despite such remarkable characteristics, polyolefin-based separators still suffer from some vital limitations that lead to poor electrochemical properties. These disadvantages include low porosity, electrolyte uptake, and inferior thermal stability [19,20,21,22]. To counteract the above-mentioned issues associated with the commercialized separators, there is a need to prepare highly porous polymer membranes that have good affinity towards liquid electrolytes and act as a gel polymer electrolyte (GPE) once impregnated with liquid electrolytes. Owing to the combined characteristics of solid and liquid electrolytes, these GPEs have displayed an improved ionic conductivity in the order of 10^−3^ S/cm while providing better safety to the lithium ion battery [23,24]. Poly(vinyl alcohol) (PVA) [25], poly(vinylidene fluoride) (PVDF) [26,27], polyacrylonitrile (PAN) [28,29], poly(methyl methacrylate) (PMMA) [30], and poly(vinylidene fluoride-hexafluropropylene) (PVDF-HFP) [31,32] are the polymer host materials primarily employed in the preparation of the GPE. PAN has some interesting features that assist it in qualifying as a good choice for preparing porous polymer membranes. Owing to the presence of a polar cyano group, PAN facilitates better transport of lithium ions. Apart from that, the stability of PAN in terms of its electrochemical and thermal behavior is sufficient. Additionally, it also does not react with the electrolytes used in the battery. In fact, PAN displays better gelation characteristics once it comes in contact with the liquid electrolyte due to the enhanced electrolyte uptake of the PAN-based membrane [33,34,35].

The introduction of ceramic materials into the polymer matrix can lead to some extraordinary characteristics owing to the mixed benefits of the two different types of materials and assists in enhancing the mechanical and thermal behaviors of the pristine polymer. Apart from that, the electrochemical features of the polymer membrane are also improved with the addition of ceramics into the polymer matrix due to the greater compatibility with the electrodes [36,37,38]. Among the different available ceramic materials, lithium aluminum titanium phosphate (LATP) has been often exploited as a solid state electrolyte owing to its high ionic conductivity of approximately 10^−4^ S/cm [39]. The improved ionic conductivity of the LATP particles can be attributed to the 3D interlocking network established due to the unique structure of LATP [40]. The polymer-LATP composite has been previously investigated as a composite electrolyte. Forsyth et al. used LATP as an additive to the polymer salt matrix of polyurethane and lithium triflate to observe how the polymer-ceramic interaction influenced the conductivity of the electrolytes [41]. In another work, Mao et al. utilized the solvent casting technique to design a PVDF-LATP composite electrolyte for application in solid-state lithium ion batteries. Enhanced electrochemical features have been previously displayed by the PVDF-LATP composite electrolyte [42]. The application of PAN-LATP composites as separators for lithium ion batteries has been studied by Zhang et al. They prepared PAN-based membranes with different concentrations of LATP, i.e., 5, 10, and 15 wt%. They observed an enhancement in the electrochemical properties of the coin cell, with a separator of 15 wt% PAN-LATP showing the best features [43]. As the 15 wt% sample displayed superior properties, it will be interesting to study how a higher concentration of LATP affects PAN–LATP membranes (PL membranes). Hence, in this work, an electrospinning technique was employed to prepare PL membranes with higher concentrations of LATP (i.e., 30 and 50 wt%). A thorough analysis of these PL membranes was performed through various structural, physical, and electrochemical characterizations. The synthesized PL membranes exhibit improved porosity and electrolyte uptake, better thermal stability, and enhanced electrochemical features, which makes them promising as separators for lithium ion batteries.

## 2. Materials and Methods

### 2.1. Fabrication of PL Membranes

Polyacrylonitrile (MW 20000) was obtained from Polysciences, Inc. (Warrington, PA, USA). Dimethylformamide (DMF) and N-Butanol were procured from Daejung Chemicals and Metals Co. Ltd. (Siheung-Si, South Korea). Electrolyte 1M LiPF_6_ in EC/DMC = 50/50 (*v*/*v*) was borrowed from Sigma Aldrich (St. Louis, MO, United States).

LATP was synthesized using a simple sol-gel method, as reported earlier by one of the authors [44]. For the fabrication of the PL membranes, first a 6 wt% PAN solution was prepared using DMF as the solvent. To this solution, different wt%s of LATP (30 and 50 wt%) were incorporated and allowed to stir at 60 °C until the LATP became completely soluble in the PAN matrix, followed by ultrasonication to prepare a homogeneous solution for electrospinning. Adopting the same procedure, PL membranes with 0 and LATP 10 wt% were prepared to report a complete trend of the PL membranes with various characteristics. A 10 mL volume of the prepared polymeric solution was removed with a syringe with a bore size 19 needle for electrospinning in a NanoNC electrospinning/spray system containing a rotating drum collector covered with aluminum foil. The voltage, flow rate, and distance between the needle and sample collector were 18 kV, 3 mL/h, and 15 cm, respectively. After the electrospinning process was completed, the sample was peeled from the aluminum foil and cut into spheres of 14 mm in diameter followed by drying in vacuum oven at 60 °C for 24 h to obtain PL membranes with thicknesses varying from 36 to 45 µm. The samples with different weight % of LATP were denoted as P-L0, P-L10, P-L30, and P-L50 for 0, 10, 30, and 50 wt%, respectively.

### 2.2. Characterization of PL Membranes

To study the morphology of the prepared PL membranes, the PL membranes were first coated with platinum, followed by field emission scattering electron microscopy (FESEM) of the membranes with a Jeol JSM-7610F (Tokyo, Japan). XRD study was done with RigakuSmartLab 3 (Tokyo, Japan) to determine the alteration in the crystallinity of the membranes with the incorporation of various weight % of LATP, whereas the Fourier transform infrared (FT-IR) spectra of the PL membranes was performed with an Agilent Cary 670. Thermogravimetric analysis of the PL membranes was performed using SETARAM instrumentation (LABSYS EVO TGA) (Caluire, France) in a nitrogen atmosphere at a scan rate of 10 °C/min to report the thermal stability of the membranes. The N-butanol soak up method was used to report the variation in porosity of the PL membranes. This method involved measuring the weight of the membranes before and after soaking them in n-butanol for two hours. After that, the following formula was employed to calculate the porosity of the PL membranes:Porosity (%) = (W_wet_**/***ρ*_b_)**/**{(W_wet_**/***ρ*_b_) + (W_dry_**/***ρ*_s_)} × 100%(1)
where W_wet_ = weight of membranes after soaking in n- butanol, W_dry_ = weight of membranes before soaking in n-butanol, V_dry_ = Volume of membrane, *ρ*_b_ = Density of n-butanol, and *ρ*_s_ = Density of separator.

To determine the electrolyte uptake of the PL membranes, the membranes were left in the dissolved liquid electrolyte, i.e.**,** LiPF6 in EC/DMC, for up to two hours. Then, the initial and final weights of membranes after electrolyte submersion were determined, which were then used in the following equation to obtain the electrolyte uptake of the PL membranes:Electrolyte Uptake (%) = {(W_wet_ − W_dry_)/W_dry_} × 100%(2)
where W_wet_ = weight of membranes after soaking in liquid electrolyte and W_dry_ = weight of membranes before soaking in liquid electrolyte.

A Nyquist impedance plot was created with the help of a Zive SP2 Electrochemical workstation (Seoul, South Korea). First, the membranes were soaked with electrolyte and then kept between the Swagelok cells for the impedance measurement, which used a frequency range of 1 Hz–1 MHz. The Nyquist plot provided the bulk resistance with which the ionic conductivity of the membranes was calculated by employing the following equation:σ = (1/R_b_) × (l/A)(3)
where σ is the ionic conductivity, R_b_ is the bulk resistance, l is the thickness of membrane, and A is the area of the membrane.

### 2.3. Electrochemical Characterization

Charge-discharge studies and the cyclic voltammetry measurements of the membranes were carried out by preparing coin cells using lithium metal foil, lithium iron phosphate (LiFePO_4_), and LiPF_6_ in EC-DMC as the anode, cathode, and electrolyte, respectively. These studies were conducted with a universal battery tester with a voltage range of 2.5 V to 4.2 V. LiFePO_4_, PVdF, and super P were employed with a ratio of 8:1:1 to prepare the LiFePO_4_ cathode in the presence of N-methyl pyrrolidone solvent. The active mass loading of the cathode was calculated to be approximately 5 mg/cm^2^.

## 3. Results and Discussion

### 3.1. Morphology and Phase Change of PL Membranes

PL membranes with varied LATP concentrations were prepared employing the facile electrospinning technique. The fabrication process is shown in Figure 1. Stirring for a long period of time followed by ultrasonication results in better homogeneity of the PL membranes. To observe the change in the morphology of the electrospun PL membranes with different amount of LATP, FESEM microscopy of the PL membranes was carried out. Figure 2 displays the FESEM micrograph of the PL membranes. The usage of the electrospinning technique assists in the formation of well aligned fibrous PL membranes, as indicated in the FESEM images. The formation of fibrous PL membranes gives rise to an improved porosity, which is shown in Table 1. The diameters of the pristine PAN fibrous membrane are found to be in the 130–260 nm range. With the incorporation of LATP particles into the PAN matrix, the polymer solution becomes more viscous than the pristine PAN solution, which results in the increase in the fiber diameters of the electrospun PL membranes. A similar increase in the fiber diameters of a P(VdF-TrFE) membrane from the introduction of Al_2_O_3_ nanoparticles has also been previously reported [45]. An increase in the viscosity with the addition of LATP particles is a major reason that it was not possible to fabricate PL membranes with higher than 50 wt% LATP. In P-L50 itself there is large amount of variation in the fiber diameters, as seen in Figure 2d, due to a greater accumulation of LATP particles, which leads the of P-L50 membrane demonstrating inferior properties when compared to P-L30 (Table 1). To report the successful inclusion of LATP ceramic particles within the PAN polymer matrix, EDS of the PL membranes was also performed using the FESEM. The presence of titanium and aluminum are seen in the EDS of the PL membranes (Appendix A), indicating fruitful incorporation of the LATP particles. The virgin PAN membrane displays only carbon and nitrogen peaks due to the nitrile group, whereas the platinum peaks in each of the PL membranes are the result of the platinum coating of the PL membranes prior to FESEM analysis. To have a clearer picture of presence of ceramic particles in PAN matrix, elemental mapping of P-L30 has been done and displayed in Appendix A. It displays all the components of PAN and LATP except the lighter element lithium.

Furthermore, XRD and FT-IR were carried out to demonstrate the interaction between the two moieties that constitute the PL membranes viz. PAN and LATP. The XRD plot shown in Figure 3a displays a characteristic peak at approximately 17° for a pristine PAN membrane [46]. Due to the crystalline characteristics of LATP, characteristic crystalline peaks of LATP start to appear alongside the peak of the pristine PAN membrane with the increase of the LATP weight %, indicating the efficacious insertion of LATP particles inside the PAN matrix. The increase in peaks upon the addition of LATP denotes the gradual transition from the polymeric to ceramic phase. P-L50 displays more crystalline LATP peaks when compared to other membranes due to presence of a higher amount of LATP. The high crystallinity of P-L50 leads to its poor electrochemical performance when compared to the P-L30 membrane, as described in Figure 5. The fine amalgamation of different moieties in the separator is very important for the electrochemical performance of the separators. FT-IR is a useful tool for demonstrating the mixing of PAN and LATP through the changes in the functional group peaks. Figure 3b shows the FT-IR of the PL membranes. The characteristic nitrile group peak for PAN is observed around 2240 cm^−1^, and the -CH_2_ stretching and bending peaks are observed around 2900 and 1400 cm^−1^, respectively. A decrease in the intensity of these peaks in the PL membranes with addition of LATP particles demonstrates the good mixing of the LATP particles within the PAN matrix. Recently, Lei et al. observed a similar type of reduction in the intensities of the FT-IR peaks of PVDF-based membranes when LATP ceramic particles were introduced into them, and ascribed it to the alteration of the PVDF bonds due to complexation between the ceramic LATP particles and PVDF [47].

### 3.2. Porosity, Electrolyte Uptake, Thermal Stability, and Ionic Conductivity of PL Membranes

Porosity is a vital criterion for battery separators, as the ability to absorb the liquid electrolyte efficiently eventually assists in achieving better electrochemical features [11]. The porosity of the PL membranes is shown in Figure 4a. The pure PAN membrane displays a porosity of approximately 67%. With the addition of LATP particles, the porosity of the PL membranes increases, as shown in Table 1. The enhancement in porosity with the addition of LATP particles can be ascribed to the increased fiber diameters [35]. The highest porosity is observed using the P-L30 membrane, i.e., 90%, after which a reduction in porosity is observed for the P-L50 membrane. This can be ascribed to the greater disparity in the fiber diameters in P-L50, as evident from the FESEM results. The improved porosity of the membranes results in a better ability to retain the electrolyte and hence a superior electrochemical performance [48].

All the PL membranes have higher porosity than that of separators (42%) employed for commercial lithium ion batteries. The electrospinning process used for PL membrane preparation in this work results in better fibrous structures with interconnected networks that in turn result in the improved porosity of the PL membranes [49]. The porosities of some reported PAN-based membranes are shown in Table 2. It is clearly visible from Table 2 that the porosity of the P-L30 membrane is the highest when compared to previously reported PAN-based membranes. The improved porosity leads to better electrochemical characteristics of the membrane [50]. The enhanced porosity is vital for the application of the membrane as a separator for lithium ion batteries, but has an adverse effect on the mechanical strength of the membranes [10,51]. The extreme stress experienced by the PL membranes was found to be approximately 10 MPa, i.e., for P-L30, as shown in Appendix A. When compared to commercialized polyethylene membranes, the tensile strength is lower but the membrane still has enough mechanical sturdiness to be employed as a separator for lithium ion batteries. 

The electrolyte uptake heavily relies on the porosity, because a higher porosity enhances the interfacial contact between the electrolyte and the separator [52]. Hence, the electrolyte uptake values for PL membranes show a trend similar to the porosity trend viz. P-L0 < P-L10 < P-L30 > P-L50. Table 1 displays the electrolyte uptake values of the PL membranes. As compared to the pristine PAN membrane, P-L10 with 10 wt% LATP shows better electrolyte uptake, which increases further with P-L30, giving a maximum electrolyte uptake value of 600%. The enhanced electrolyte uptake and porosity of the P-L30 membrane improves the lithium insertion and extraction during the charge-discharge cycle and hence results in better cyclic stability [50]. However, increasing the LATP concentration to 50 wt% leads to a reduction in the electrolyte uptake owing to a greater aggregation of LATP in the polymer matrix. Recently, Thomas et al. described similar electrolyte uptake behavior in which, with an increase in the amount of Al_2_O_3_, the electrolyte uptake of a P(VdF-TrFE) membrane increases but, in case of an even higher quantity of Al_2_O_3_, the electrolyte uptake values decreased [45]. 

Another major concern for separators is thermal stability. The safety of the battery depends on the improved thermal stability of the separator. Highly thermally stable separators can resist high temperatures and can avert shrinkage at elevated temperatures, which impacts the performance of the lithium ion battery. The thermal stability of the PL membranes was investigated through the TGA measurements depicted in Figure 4b, in which the samples were exposed to a temperature range starting from room temperature to 800 °C at a scan rate of 10 °C/min. A major weight loss occurs between 290 °C and 300 °C due to the melting of the polymer host, i.e., PAN, after which no major weight loss occurs, and only the membrane decomposes. Still, after 800 °C, the residual weight percents of the PL membranes are 16%, 59%, 63%, and 35% for P-L0, P-L10, P-L30, and P-L50, respectively, whereas a reduction in weight of more than 90% is observed in the case of commercially used polyethylene and polypropylene separators by 600 °C [55,56]. The enhanced residual weight % of PL membranes compared to commercial polypropylene separator and pristine PAN membrane, i.e., P-L0, can be ascribed to the fine thermal properties of the added LATP particles [47]. As is evident, the P-L30 membrane is more thermally stable when compared to the other samples, indicating that the P-L30 membrane is the best sample from the lot, which is also well corroborated by the porosity and electrolyte uptake results. Additionally, digital images of the celgard separator and P-L30 membrane are depicted in Appendix A before and after heat treatment at 170 °C for 1 h. After 1 h, the celgard separator was completely melted down, whereas in case of P-L30 the color changed from white to yellow but there was little thermal shrinkage (~14%) of P-L30 membrane, indicating the enhanced thermal stability of PL membrane compared to commercial celgard separator.

Figure 4c shows the Nyquist impedance plots of the PL membranes soaked in liquid electrolyte, i.e., LiPF6 in EC-DMC. The point at which the curves of the PL membranes touch the real impedance axis is considered to be the bulk resistance, Rb, of the membranes. Taking Rb, the ionic conductivity of the membranes was determined using Equation (3). The calculated ionic conductivity values are listed in Table 1. The P-L0 membrane, i.e., the membrane without LATP, demonstrated an ionic conductivity of 0.22 mS/cm, but increases in ionic conductivity values have been observed in the incorporation of LATP into the membrane, which can be ascribed to the higher ionic conductivity of the LATP particles [40,43]. The ionic conductivity values of P-L10, P-L30, and P-L50 are 1.4 mS/cm, 1.7 mS/cm, and 0.89 mS/cm, respectively. The enhanced electrolyte uptake and porosity of the P-L30 membrane improves its ionic conductivity above all the other PL membranes [57,58]. The improved ionic conductivity of P-L30 membrane is further corroborated through the cyclic voltammetry measurements (Appendix A), in which the difference between the anodic and cathodic peak is found to be the lowest when using the P-L30 membrane (i.e., 293 mV), indicating high lithium ion transportation in the P-L30 case. More aggregation of ceramic particles gives rise to less electrolyte uptake of P-L50, which leads to display of low ionic conductivity of P-L50 compared to P-L10 and P-L30.

### 3.3. Electrochemical Investigation of PL Membranes

To test the efficiency of the PL membranes for use in a lithium ion battery, the PL membranes were assembled in coin cells with a lithium metal anode and a LIFePO4 cathode, using LiPF6 in EC-DMC as the liquid electrolyte. The initial charge–discharge profiles of the cells with PL membranes are displayed in Figure 5a. It can be observed that cells with a pristine PAN membrane without any LATP have a discharge capacity of approximately 134 mAh/g at 0.5C rate, which improves with the inclusion of LATP. A 30 wt% inclusion shows the highest discharge capacity at 158 mAh/g. The elevated ionic conductivity of the P-L30 membrane results in the improved charge–discharge behavior of the coin cell with the P-L30 membrane and hence overall results in the improved capacity of the coin cell [59]. Due to a reduction in porosity and electrolyte uptake, the coin cell with PL membrane with 50 wt% LATP shows a reduced capacity when compared to the coin cell with a P-L30 membrane. The charge–discharge curves of the Celgard separator at 0.5C rate are also included with the charge–discharge curves PL membranes, and it can be seen that there is very little difference in the specific capacity value between the coin cells with a Celgard separator and with only a PAN membrane, but the capacity of a coin cell with the Celgard separator is much lower when compared to coin cells with LATP containing PL membranes. The enhanced capacity of the PL membranes can be ascribed to the better electrolyte retaining ability, which comes from their fibrous structure. This result demonstrates that PL membranes containing LATP could be a better option than commercial separators. 

Figure 5b shows the rate capability of the PL membranes by varying the C-rate from 0.5C to 4C. Due to the lithium insertion/extraction mechanism of the LIFePO4 cathode, a reduction in the discharge capacities of the PL membranes is observed as the C-Rate increases [60]. Still, the P-L30 membrane maintains 56% of its initial capacity at a rate of 4C, whereas the capacity retention of P-L0, P-L10, and P-L50 is 26%, 49%, and 39%, respectively. This indicates that the high electrolyte uptake of the P-L30 membrane contributes to efficient lithium ion intercalation and deintercalation, even at a rate of 4C. All the PL membranes display efficacious reversibility, as the capacity shows little change when the rate returns from 4C to a smaller C rate, i.e., 0.5C. The greater reversibility of the PL membranes can be ascribed to the homogeneous mixing of the LATP within the polymer matrix, as evident from the FTIR studies [61].

Since the coin cell with the P-L30 membrane displays the highest discharge capacity, the long-term stability of the P-L30 membrane for use in a lithium ion battery was tested. This was performed by measuring the cycling performance of P-L30 membrane containing coin cell at a 0.5C rate for 200 cycles, shown in Figure 5c. It was found that even after undergoing 200 charge-discharge cycles, the coin cell using the P-L30 membrane retained 87% of its initial discharge capacity. The enhanced cycling stability of the coin cell with the PL-30 membrane can be attributed to higher porosity of the P-L30 membrane, which restricts the evolution of lithium dendrites during repeated cycling [62]. Further, the enhanced electrolyte uptake and increased ionic conductivity also plays a crucial role in the cycling stability of the P-L30 membrane. Appendix A displays the FESEM image of the P-L30 membrane after undergoing 200 cycles of charge and discharge. Even after 200 cycles, there is no major changes in the fiber diameters; only deposition of particles was observed between the fibers, which may have been due to the LFP cathode. Such type of particle deposit was also observed by Lee et al. in the PE separators after undergoing 100 cycles of charge and discharge [63]. The coulombic efficiency changes of the coin cell containing the P-L30 membrane are also displayed in Figure 5c. The initial charge–discharge cycle at a rate of 0.5C shows a coulombic efficiency of 99.93%, and after 200 cycles, it still maintains a high coulombic efficiency of 96.5%, indicating the efficacy of using the P-L30 membrane as a lithium ion battery separator. 

## 4. Conclusions

An electrospinning procedure has been employed to prepare PAN-based membranes with high concentrations of LATP particles, i.e., 30 and 50 wt%. To have a complete variation trend demonstrating different properties, membranes with 0 and 10 wt% were also prepared. The FESEM results showed that the pure PAN membrane was a well-aligned fibrous membrane, but upon increasing the LATP concentration, irregularities in the fiber diameters were noticed. XRD and FT-IR measurements showed the fine interaction between the PAN and LATP particles. Due to a higher porosity, the P-L30 membrane displayed improved characteristics such as better electrolyte uptake, greater thermal stability, and an augmented ionic conductivity of 1.7 mS/cm, results that were superior to those of the other PL membranes tested. LiFePO4/Li-based coin cells prepared with a P-L30 membrane exhibited an enhanced discharge capacity of 158 mAh/g at 0.5C rate and displayed good capacity retention with a higher C-rate. Furthermore, only a 13% reduction in the capacity was observed while cycling the coin cell with the P-L30 membrane for 200 cycles at a 0.5C rate, and an improved coulombic efficiency of 96.5% was retained after charging and discharging the P-L30-based coin cell for 200 cycles at a rate of 0.5C. Hence, the prepared PL membranes are promising candidates for lithium ion battery applications.

## Figures and Tables

**Figure 1 nanomaterials-09-01581-f001:**
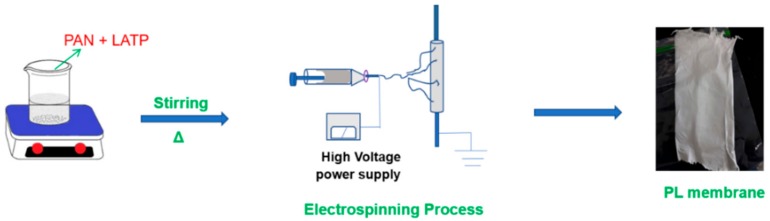
Preparation process of polyacrylonitrile (PAN)–lithium aluminum titanium phosphate (LATP) membranes (PL membranes).

**Figure 2 nanomaterials-09-01581-f002:**
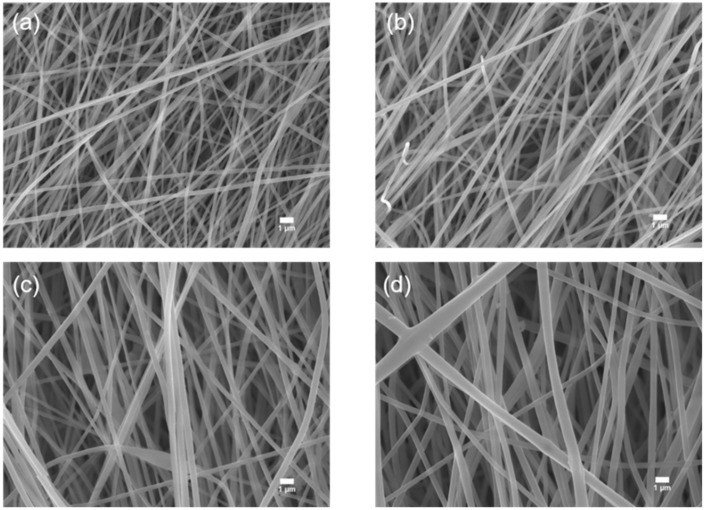
Field emission scattering electron microscopy (FESEM) of (**a**) P-L0; (**b**) P-L10; (**c**) P-L30; and (**d**) P-L50.

**Figure 3 nanomaterials-09-01581-f003:**
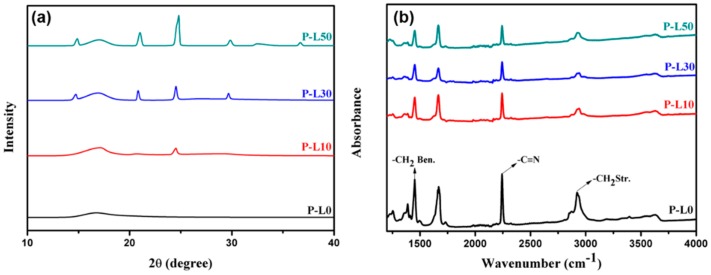
(**a**) XRD and (**b**) Fourier transform infrared (FT-IR) spectra of PL membranes.

**Figure 4 nanomaterials-09-01581-f004:**
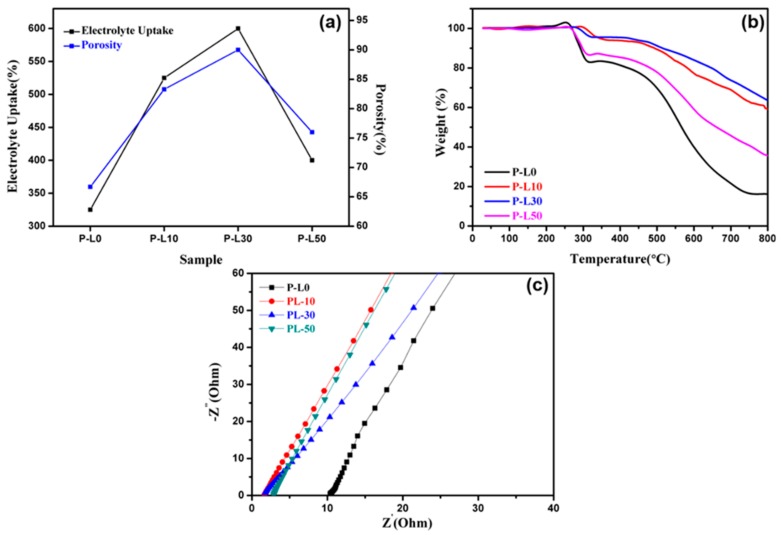
(**a**) Porosity and electrolyte uptake; (**b**) Thermogravimetric Analysis (TGA); and (**c**) Nyquist impedance plot of PL membranes.

**Figure 5 nanomaterials-09-01581-f005:**
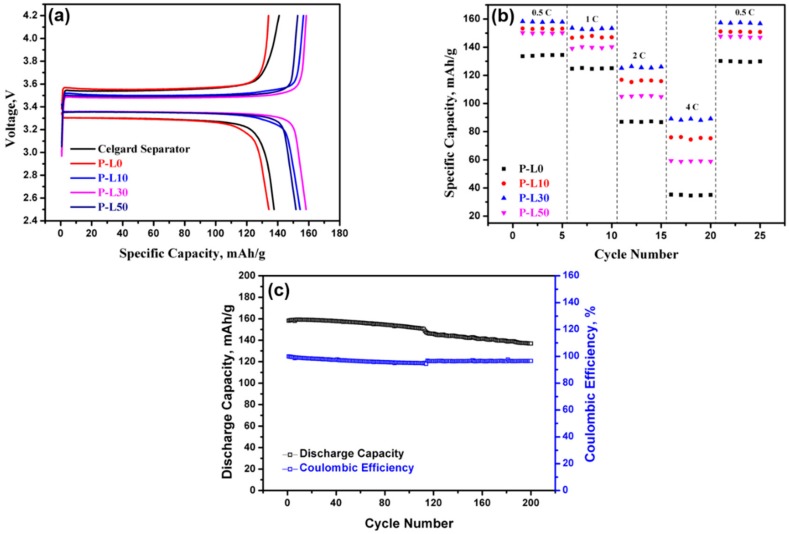
(**a**) Initial charge-discharge curve with 0.5C rate of PL membranes, (**b**) rate capability of PL membranes, and (**c**) cyclic stability and efficiency of P-L30.

**Table 1 nanomaterials-09-01581-t001:** Porosity, electrolyte uptake, and ionic conductivity values of PL membranes.

Sample	Porosity (%)	Electrolyte Uptake (%)	Ionic Conductivity (mS/cm)
P-L0	66.7	325	0.22
P-L10	83.3	525	1.4
P-L30	90	600	1.7
P-L50	76	400	0.89

**Table 2 nanomaterials-09-01581-t002:** Porosity of PAN-based membranes.

PAN Based Separator Works	Porosity (%)	Reference
Electrospun PAN membranes	68	[53]
SiO_2_/PAN	77	[54]
Lignin/PAN	74	[35]
PAN/PI	87	[50]
PAN-LATP (P-L30)	90	This Work

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
