# Peer review of "Preparation of Highly Porous PAN-LATP Membranes as Separators for Lithium Ion Batteries"

_nanomaterials, 2019, doi:10.3390/nano9111581_

Round 1
Reviewer 1 Report
The manuscript properly describes the background and the purpose of this study and the results are clearly presented showing improved properties of the developed material. Although it may be sufficient as a technical report, it lacks scientific insight why the certain composition of PAN-LATP gives the better performance than other composition.
In the polymer composites it would be crucial to analyze spatial distribution of the polymers and solid materials. It is difficult to identify the presence of the solid particles within the fibers in the FE-SEM images in Figure 2 and it seems that the ceramic particles are homogeneously dispersed in the fibers. The authors show EDS of the membranes but EDS mapping would help the mutual distribution of the polymers and ceramics in the fibers. In FT-IR spectra in Figure 3(b) the peak at around 1650 cm-1 are left unassigned. The FT-IR spectra of PAN do not show any sharp peak around this position and it would be difficult to suppose that LATP can have a sharp peak at this position. I suppose this peak and the peaks of CH2 stretching at around 2900 cm-1 and the small peaks at around 1400 cm-1 are due to DMF remaining in the materials. The ordering of the relative intensities of the peak at 1650 cm-1 against the peak at 2250 cm-1 which can unambiguously assigned to PAN is L30 < L10 < L50 <L0, which is just the opposite of the porosity and the electrolyte uptake. I suspect that the changes in the porosity and the uptake with the composition is due to the amount of the remaining DMF in the materials and its complete removal would change their relative performance.Author Response
Review 1:
The manuscript properly describes the background and the purpose of this study and the results are clearly presented showing improved properties of the developed material. Although it may be sufficient as a technical report, it lacks scientific insight why the certain composition of PAN-LATP gives the better performance than other composition.
Authors would like to thank the reviewer’s comments on our manuscript.
Response : Thank you for your comment. The PAN LATP composition containing 30% of LATP display better performance due to enhancement in porosity and electrolyte uptake which is explained thoroughly in the results and discussion with the help of different references.
In the polymer composites it would be crucial to analyze spatial distribution of the polymers and solid materials. It is difficult to identify the presence of the solid particles within the fibers in the FE-SEM images in Figure 2 and it seems that the ceramic particles are homogeneously dispersed in the fibers. The authors show EDS of the membranes but EDS mapping would help the mutual distribution of the polymers and ceramics in the fibers. In FT-IR spectra in Figure 3(b) the peak at around 1650 cm-1 are left unassigned. The FT-IR spectra of PAN do not show any sharp peak around this position and it would be difficult to suppose that LATP can have a sharp peak at this position. I suppose this peak and the peaks of CH2 stretching at around 2900 cm-1 and the small peaks at around 1400 cm-1 are due to DMF remaining in the materials. The ordering of the relative intensities of the peak at 1650 cm-1 against the peak at 2250 cm-1 which can unambiguously assigned to PAN is L30 < L10 < L50 <L0, which is just the opposite of the porosity and the electrolyte uptake. I suspect that the changes in the porosity and the uptake with the composition is due to the amount of the remaining DMF in the materials and its complete removal would change their relative performance.
Response: Thank you for your comment. To have a better outlook of polymer and ceramic distribution elemental mapping of P-L30 membrane has been incorporated in the supplementary information (figure S2) which indicates the presence of all components of both PAN and LATP ceramics.
DMF has a formula (CH3)2NC(O)H and it does not have any -CH2 bonds. The nitrile peak at around 2240 cm-1 and the -CH2 bending and stretching peaks at 1400 and 2900 cm-1 respectively are characterised peaks of the PAN which seems to decrease with incorporation of LATP, indicating well mixing of PAN and LATP.
Reviewer 2 Report
This work describes the preparation of PAN/LATP membranes via electrospinning for use as separators in lithium ion batteries.
The incorporation of the LATP ceramic particles (5,10, 15 wt%) in the PAN matrix for the preparation of separators for lithium ion batteries has already been reported in the literature, thus there is no novelty in this work. The authors prepared membranes with LATP wt% 10, 30 and 50%. The prepared membranes were characterized in terms of morphology, porosity, thermal stability, electrochemical characterization and FT-IR, XRD techniques. The interpretation of the data is quite sufficient.
To summarize, it is a well-written manuscript, however,the novelty of this work is poor. I recommend this work to be published in Nanomaterials after consideration of the following issues
1. In the references, there are missing 1-2 review articles related to separators for lithium ion batteries that should be added.
2. Regarding the thermal stability of the prepared membranes, the shrinkage test should be conducted not only at 80 for 1h, but at higher temperatures up to 170C.
3. In the experimental part, should be added a paragraph regarding the chemicals that have been used
Author Response
Review 2:
This work describes the preparation of PAN/LATP membranes via electrospinning for use as separators in lithium ion batteries.
The incorporation of the LATP ceramic particles (5,10, 15 wt%) in the PAN matrix for the preparation of separators for lithium ion batteries has already been reported in the literature, thus there is no novelty in this work. The authors prepared membranes with LATP wt% 10, 30 and 50%. The prepared membranes were characterized in terms of morphology, porosity, thermal stability, electrochemical characterization and FT-IR, XRD techniques. The interpretation of the data is quite sufficient.
Authors would like to thank the reviewer’s comments on our manuscript.
To summarize, it is a well-written manuscript, however, the novelty of this work is poor. I recommend this work to be published in Nanomaterials after consideration of the following issues
Response: Thank you for appreciating the interpretation of data and writing of the manuscript.
In the references, there are missing 1-2 review articles related to separators for lithium ion batteries that should be added.
Response: Thank you for the suggestion. Two recent review articles related to lithium ion battery separators i.e. reference 15 and 16 has been incorporated in the manuscript.
Regarding the thermal stability of the prepared membranes, the shrinkage test should be conducted not only at 80 for 1h, but at higher temperatures up to 170C.
Response: Thank you for the suggestion. The thermal shrinkage test of P-L30 membrane and celgard separator has been included in supplementary inforation (figure S3) and the required explanation was incorporated in the manuscript and was highlighted. (Page-8 Line - 305-309)
In the experimental part, should be added a paragraph regarding the chemicals that have been used
Response : Thank you for the suggestion. All the chemicals used in current experiment and the suppliers were mentioned in the manuscript and was highlighted. (Page-3 Line – 100-102)
Round 2
Reviewer 1 Report
After reading the revised manuscript and some pieces of cited literature on the recent development of the related compounds, I understood that the results of the FESEM, EDX, XRD, and FT-IR are consistent with the performance of the PAN-LATP which changes with the added amount of LATP. In my opinion, the effect of LATP still needs further clarification based on the analysis of the form of the LATP in PAN-LATP, i.e., whether they are incorporated in the fibers or dispersed on the surface of the fibers, or only segregated as crystalline particles as observed in XRD profiles and EDS mapping images in Figure S2, as they should alter the surface states of the fibers and eventually affect porosity as well as electrolyte uptake. However, this issue is out of the scope of this manuscript and it is acceptable with following minor corrections.
1) In the abstract, the name of LATP is denoted as "Lithium titanium aluminum phosphate", which is inconsistent with the abbreviation.
2) I commented that the peak at 1650 cm-1 which was left unassigned in the FT-IR might be due to the remaining DMF. This point was not answered in the reply from the authors.
Author Response
Review 1:
After reading the revised manuscript and some pieces of cited literature on the recent development of the related compounds, I understood that the results of the FESEM, EDX, XRD, and FT-IR are consistent with the performance of the PAN-LATP which changes with the added amount of LATP. In my opinion, the effect of LATP still needs further clarification based on the analysis of the form of the LATP in PAN-LATP, i.e., whether they are incorporated in the fibers or dispersed on the surface of the fibers, or only segregated as crystalline particles as observed in XRD profiles and EDS mapping images in Figure S2, as they should alter the surface states of the fibers and eventually affect porosity as well as electrolyte uptake. However, this issue is out of the scope of this manuscript and it is acceptable with following minor corrections.
1) In the abstract, the name of LATP is denoted as "Lithium titanium aluminum phosphate", which is inconsistent with the abbreviation.
Response: Thank you for the suggestion. Required correction has been done in the abstract highlighted in yellow colour.
2) I commented that the peak at 1650 cm-1 which was left unassigned in the FT-IR might be due to the remaining DMF. This point was not answered in the reply from the authors.
Response: The peak around 1650 cm-1 is due to -C=O stretching of the remaining DMF. There is some issue with the peak intensity which has been resolved now as you can see in figure-3.